# Sex Reversal Induced by Dietary Supplementation with 17α-Methyltestosterone during the Critical Period of Sex Differentiation in Oriental River Prawn (*Macrobrachium nipponense*)

**DOI:** 10.3390/ani13081369

**Published:** 2023-04-17

**Authors:** Pengfei Cai, Huwei Yuan, Zijian Gao, Peter Daka, Hui Qiao, Wenyi Zhang, Sufei Jiang, Yiwei Xiong, Yongsheng Gong, Yan Wu, Shubo Jin, Hongtuo Fu

**Affiliations:** 1Wuxi Fisheries College, Nanjing Agricultural University, Wuxi 214081, China; ckgg5436@126.com (P.C.); yuan08102021@126.com (H.Y.); gaozijiangenomics@163.com (Z.G.); peterdaka1@yahoo.com (P.D.); 2Key Laboratory of Freshwater Fisheries and Germplasm Resources Utilization, Ministry of Agriculture and Rural Affairs, Freshwater Fisheries Research Center, Chinese Academy of Fishery Sciences, Wuxi 214081, China; qiaoh@ffrc.cn (H.Q.); zhangwy@ffrc.cn (W.Z.); jiangsf@ffrc.cn (S.J.); xiongyw@ffrc.cn (Y.X.); gongys@ffrc.cn (Y.G.); wuy@ffrc.cn (Y.W.)

**Keywords:** 17α-methyltestosterone, gonadal development, histological observations, *Macrobrachium nipponense*, neo-males, sex ratio, sex reversal

## Abstract

**Simple Summary:**

The steroid 17α-methyltestosterone (MT) inhibits ovarian function and is often used to induce sex reversal artificially in vertebrates. In oriental river prawn, sex reversal through vertebrate sex hormones can be observed. Neo-males (sex-reversed female prawns) were maintained by exogenous androgen, and over-reliance led to slow testis growth, small body size, and low growth rate, but sperm was still produced. In female prawns, MT inhibited ovary development and promoted growth.

**Abstract:**

The steroid 17α-methyltestosterone (MT) inhibits ovarian function and is often used to induce sex reversal artificially in vertebrates. In the present study, different concentrations of MT were added as dietary supplementation, and the effects on sex ratio, growth, and gonadal development were examined. After 40 days, the sex ratio (male:female) in each group increased at different degrees with 50 (1.36:1), 100 (1.57:1), and 200 (2.61:1) mg/kg MT, and neo-males with testis–ovary coexistence were observed in the 200 mg/kg MT group. Furthermore, 50 and 100 mg/kg MT could induce female reversion in neo-males. Histologically, the development of the testes in experimental groups was slower, but the ovaries of the experimental and control groups had similar developmental rates. The expression levels of *DMRT11E*, *Foxl2*, and *SoxE1* in males at 200 mg/kg MT were 8.65-, 3.75-, and 3.45-fold greater than those of the control group. In crustaceans, sex reversal through vertebrate sex hormones can be observed. Neo-males (sex-reversed female prawns) were maintained by exogenous androgen, and over-reliance led to slow testis growth, small body size, and low growth rate, but sperm was still produced. In female prawns, MT inhibited ovary development and promoted growth.

## 1. Introduction

The oriental river prawn (*Macrobrachium nipponense*; Crustacea; Decapoda; Palaemonidae) is farmed in China on a large scale [1]. Although the annual aquaculture production was about 224,413 tons in 2021 and the output value was more than 20 billion, some problems exist in its development. The male has a faster growth rate and better disease resistance than the female [2]. Therefore, studies on the mechanisms of sex differentiation and reliable monosex cultivation techniques have important practical value. Artificially induced sex reversal is a first step, and its success depends on the critical period of sex differentiation [3].

Sex differentiation is the process by which undifferentiated gonads with bidirectional potential are programmed to develop into spermatophores or ovaries, and develop secondary sexual traits through a series of events [4]. The androgenic glands are a unique endocrine glandular tissue in crustaceans that play a key role in sexual differentiation to maleness, including the development of the testes and male sexual characteristics [5]. Androgenic gland hormone (AGH) and its relative producers are recognized as important factors in sex differentiation and determination, and their functions have been widely studied in many crustaceans [6]. Previous studies have shown that in oriental river prawns, the glands begin to develop at PL10 (PL: post-larvae developmental stage) and sexual differentiation is complete at PL25 with the emergence of physiological males and females, although the secondary sexual traits are not apparent [7].

Steroid hormones, mainly found in the testes, ovaries, hepatopancreas, and hemolymph, play a significant role in the regulation of gonadal development, sex determination, and growth in aquatic species through interactions with endocrine factors [8]. 17α-Methyltestosterone (MT), a type of steroid that inhibits ovarian function and endometrial growth, is one of the most studied hormones in animals. Several studies have reported that MT can stimulate the sexual development of males in vertebrates [9]. The addition of MT to feed is the easiest and most convenient mode of obtaining male individuals through sexual differentiation [10]. A recent study reported that sex reversal is possible in under-yearlings of the Malabar grouper (*E*. *malabaricus*) using synthetic androgen during the period of ovarian differentiation [11]. Female white grouper (*E. aeneus*) can be sex-reversed to males that produce sperm and could be used to establish egg, larvae, and juvenile production in aquaculture [12]. The application of MT before the labile period of sexual differentiation can induce complete sex reversal in mandarin fish (*S. chuatsi*) [13]. Crustaceans are lower aquatic animals and may be more susceptible to sex reversal when testosterone is present during early development [14]. However, there are no successful cases of sex reversal induced by vertebrate sexual hormones. A report on juvenile Chinese vannamei shrimp (*L. vannamei*) showed that soaking with MT can increase the percentage of male prawns by 66.7% [15]. A similar study in freshwater prawn (*M. rosenbergii*) found that sex reversal with testosterone was difficult after the developmentally sensitive period that causes sex differentiation had passed [16]. Therefore, selecting the critical period of sex differentiation in oriental river prawn to administer MT may be an important breakthrough in inducing sex reversal.

Sperm gelatinase (SG) may play an important role in the regulation of sperm motility [17] (hyperactivation), acrosome reaction [18], sperm–egg fusion [19], and many other reproductive functions. *Mn-SG* is specifically expressed in the testes of oriental river prawns, and the expression level gradually increases with the degree of testis development. The level of SG decreases after RNA interference (RNAi)-induced knockdown of *Mn-SG*. The insulin-like androgenic gland hormone (*IAG)* encodes and secretes many androgens that play key roles in male differentiation in crustaceans [20,21]. In oriental river prawns, the expression of androgen-related genes is decreased in the androgenic gland (AG) by RNAi-induced knockdown of *IAG* [22]. The rapid production and deposition of the yolk protein precursor vitellogenin (VG), which is used by embryos and larvae during early development, determine oocyte growth and gonadal maturation [23,24]. During vitellogenesis, the VG receptor (VGR) secretes VG into the hemolymph receptor-mediated endocytosis and incorporates VG into the developing oocyte, where it is eventually converted into yolk protein [25]. In oriental river prawn, RNAi-induced knockdown of *VG* and *VGR* inhibit ovarian maturation, and *VG* RNAi downregulates *Mn-VGR* gene expression in the ovary space [26,27].

The present study aimed to determine whether dietary supplementation with different concentrations of MT could induce sex reversal in the oriental river prawn. The effects of MT supplementation on growth and gonadal development were assessed through sex ratio, histological observations, and qPCR analysis. In addition, the genes that may contribute to the sexual differentiation of the oriental river prawn and the interaction of these genes were explored. The present study demonstrated that vertebrate sex hormones could induce sex reversal in crustaceans, and explored MT concentrations and periods that induce sex reversal in the oriental river prawn. The results of the study provided evidence that MT regulated sex differentiation and led to sex reversal in juvenile oriental river prawns.

## 2. Materials and Methods

### 2.1. Experimental Prawns

Healthy pregnant female oriental river prawns (body weight = 4.02 ± 0.55 g) were obtained from Taihu Lake (Wuxi, China; 120°13′44″ E, 31°28′22″ N) and maintained in a 500 L tank with a dissolved oxygen content of ≥6 mg/L at room temperature (28 ± 1 °C). The juvenile prawns hatched from these females were also cultured under the same conditions. After metamorphosis, the larvae were fed with Artemia until body weight reached 0.0434 ± 0.0002 g (Table 1). During this period, the gonads of the oriental river prawns were completely developed, and secondary sexual characteristics begin to appear gradually. Three replications of each experimental group were made and contained 200 juvenile prawns. The diets with different concentrations of MT were fed two times (8 a.m. and 8 p.m.) per day at 2% of total body weight [28].

### 2.2. Dietary Preparation

The diets used in this study were commercial prawn feed produced by Guangzhou Liyang Aquatic Products Co., Ltd. (Guangzhou, China). The commercial diet is mainly composed of crude protein, fish meal, shrimp meal, squid meal, starch, soybean meal, ash, canola meal, soybean protein concentrate, and crude lipid [29]. The MT (CAS number: 58-18-4, purity: 97.41%) was purchased from Beijing Solarbio Technology Co., Ltd. (Beijing, China).

The method of dissolving MT into the diets is described as follows [30]: MT was dissolved in 95% ethanol to prepare a stock solution at a concentration of 50 mg/mL and then diluted into concentrations of 5, 10, and 20 mg/mL. Different concentrations of ethanol were then evenly sprayed on the feed (1 mL ethanol per 10 g diet) and stirred with a glass stick for at least 3 min. The diets were then placed under a ventilated laboratory hood and left in the shade for 15 min. The treated diets were added to 15 mL test tubes and placed in a refrigerator at 0 °C to evaporate the remaining alcohol naturally. The diets were left until the color was the same as normal diets and without the smell of alcohol. The control diet was prepared in the same manner using 95% ethanol without the MT. The diets were stored at 4 °C until feeding.

### 2.3. Experimental Design

At 25 days after metamorphosis, the juvenile oriental river prawns were fed on diets with different concentrations of MT for 40 days. The experimental design is shown in Figure 1. The start of the MT treatment was set as day 0 (PL25), and subsequent sampling stages were denoted as days 10, 20, 30, and 40.

### 2.4. Sex Ratio Statistics

The prawns in the control and experimental groups were randomly selected from more than 90 individuals at 10, 20, 30, and 40 days to determine the sex ratio. Each group had at least three replicates. The distinctions between the secondary sexual characteristics of male and female prawns are shown in Table 2 [31].

### 2.5. Measurement of the Growth Traits

The average body weights of juvenile oriental river prawns were determined at the beginning of the experiment on day 0 (initial mean weight) and 40 days (final mean weight) after the MT treatment. The body weights of 50 randomly selected prawns from each replicate were measured.

The weight growth rate (*WGR*) was determined as follows:
WGR=(Wt−W0)/W0×100%


The specific growth rate (*SGR*) was determined as follows:
SGR=(In Wt−In W0)/t×100%

where *Wt* is the average body weight 40 days after treatment with MT, *W*0 is the average body weight before treatment with MT, and *t* is the total days of culture [32].

### 2.6. Histological Observations of Testicular Development

Male and female *M. nipponense* were separated after being treated with different concentrations of MT and stained with hematoxylin and eosin (HE) to study the histological changes in the testis and ovary. After 40 days, samples of male and female prawns from the control and 200 mg/kg MT group were mounted on slides and stained with HE, and prepared as described in previous studies [33]. Observations took place using a stereo microscope (SZX16; Olympus Corporation, Tokyo, Japan). Comparative labeling was performed with various cell types based on cell morphology [7].

### 2.7. The qPCR Analysis

After treatment with different concentrations of MT, qPCR analysis was used to measure the expression levels of eukaryotic translation initiation factor 5A (*EIF*, GenBank accession number: MH540106), Mn-insulin-like androgenic gland hormone (*Mn-IAG,* GenBank accession number: KC460325), Mn-sperm gelatinase (*Mn-SG*, GenBank accession number: KF831095), Mn-vitellogenin (*Mn-Vg*, GenBank accession number: KJ768657), Mn-vitellogenin receptor (*Mn-Vgr*, GenBank accession number: KJ768658), the doublesex and mab-3 related transcription factor (*Mn-DMRT*, GenBank accession number: MH636338), forkhead box protein L2-like (*Mn-Foxl2*, GenBank accession number: KM821275), and subgroup E within the Sox family of transcription factors (*Mn-SoxE*, GenBank accession number: MN271693).

Juvenile male and female prawns treated with different concentrations of MT were separated and immediately frozen in liquid nitrogen to prevent RNA degradation. Three male and three female prawns in each replicate were randomly selected for RNA extraction from the whole sample. RNA was extracted using RNAiso Plus Reagent (Takara Bio. Inc., Shiga, Japan). The concentration of RNA was quantified by using a Bio-Photometer (Eppendorf, Hamburg, Germany) and 1 μg of total RNA from each sample was reverse-transcribed into cDNA using a Prime Script RT reagent Kit (Takara Bio. Inc.) following the manufacturers’ protocols. The expression levels of the genes tested in each sample type were identified by qPCR using Ultra SYBR Mixture (CWBio, Beijing, China). The qPCR amplifications were carried out in a total volume of 25 μL, containing 1 μL cDNA (50 ng), 10 μL SsoFast EvaGreen Supermix (Bio-Rad, Washington, USA), 0.5 μL of gene-specific forward and reverse primers (listed in Table 3), and 13 μL of DEPC water. The reaction mixture was initially incubated at 95 °C for 30 s to activate the Hot Start Taq DNA polymerase, followed by 40 cycles at 95 °C for 10 s and 60 °C for 1 s. A melting curve was conducted at the end of the qPCR reaction at 65–95 °C (in 0.5 °C increments) for 10 s. At least three replicate qPCRs were performed per sample, in each concentration gradient of female prawns n ≥ 3 and male prawns n ≥ 3, and EIF was used as the reference gene [34]. The relative copy number of genes between different concentrations was calculated using the 2^−ΔΔCT^ method [35].

### 2.8. Statistical Analysis

The statistical analyses were all conducted using IBM SPSS Statistics for Windows, version 23.0. (IBM Corporation, Armonk, NY, USA). The significant differences between groups were determined by one-way ANOVA, followed by the least significant difference and Tukey’s test. In addition, to determine if the effect was linear and/or quadratic, a follow-up trend analysis using orthogonal polynomial contrasts was performed [36]. Quantitative data are expressed as mean ± standard error of the mean (SEM). Probability (*p*) values < 0.05 were considered statistically significant.

## 3. Results

### 3.1. Effects of MT Concentration on the Sex Ratio of Juvenile Prawns

The results for the sex ratio (male:female) at four different concentrations of MT over various lengths of time are presented in Figure 2. As shown in Figure 2A, throughout the culture phase the sex ratio of the control group was maintained around 1. In Figure 2B, the sex ratio gradually decreases with increasing culture time. Compared with the control group, the ratio was significantly higher at 10 and 20 days (*p* < 0.05). However, the sex ratio at 30 and 40 days were 1.44 and 1.36. respectively, and were not significantly different. The sex ratio showed a similar trend of decline in Figure 2C, at 40 days it was 1.57. Compared to the 50 and 100 mg/kg MT experimental groups (Figure 2D), the sex ratio was high and stable in the 200 mg/kg MT experimental group, with significant differences between the four concentrations (*p* < 0.05).

### 3.2. Histological Observations of the Testis and Ovary

As shown in Figure 3A, after 40 days of culture, the normal developing male prawn was in the spermatid phase, with primary and second spermatocytes in the testis and a large number of sperm observed in the vas deferens. Normally developed sperm were observed, as shown in Figure 3E, but some male prawns had fewer sperm compared to the control shown in Figure 3C. Figure 3B,D show that the ovaries of the control and experimental groups are at the same stage of development, with many oogonia and primary oocytes observed and no yolk granule accumulation. The testis–ovary results are shown in Figure 3E. Follicle cells and cytoplasmic membrane can be seen in the ovarian cavity. The testes are filled with primary and secondary spermatocytes, but no mature sperm.

### 3.3. Effects of MT Concentration on Growth Traits of Juvenile Prawns

Effects of different concentrations of MT on growth traits of male and female juvenile prawns are described in Table 4. In males (Table 4), after 40 days of treatment with the gradual increase in MT concentration, the WGR and SGR of males decreased gradually, and there were significant differences between the 50, 100, and 200 mg/kg MT groups and the control group (*p* < 0.05), but there was no difference within the three treatment groups. The WGR and SGR of 200 mg/kg MT were 1628.87 ± 77.43 and 5.86 ± 0.05, respectively, which were less than 25% of the values of the control group. The FMW, WGR, and SGR of male prawns significantly decreased linearly with an increase in dietary MT levels (*p* < 0.05).

In females (Table 4), the WGR and SGR increased gradually with the increase in MT concentration after 40 days of treatment. The WGR and SGR of the 200 mg/kg MT group were 1549.79 ± 72.32 and 5.59 ± 0.09, respectively, which was more than 30% greater than that of the control group (*p* < 0.05). The FMW, WGR, and SGR of female prawns significantly increased linearly with the increase in dietary MT levels (*p* < 0.05). Moreover, 40 days after 200 mg/kg MT was administered to juvenile prawns, the WGR and SGR of females and males tended to be the same and were not significantly different in SPSS analysis.

### 3.4. Analysis of Male-Specific and Female-Specific Genes by qRT-PCR

All the samples measured were whole prawns that came from three separate individuals in the same group. As shown in Figure 4, *Mn-IAG* and *Mn-SG* mRNA levels in males were measured at 40 days after different concentrations of MT. An increase in MT concentration dramatically decreased the expression levels of *Mn-IAG*. The expression characterization of *Mn-IAG* in experimental groups was lower than the control (*p* < 0.05). Moreover, the expression level of *Mn-IAG* in the control group was 17.9-fold greater than that in the 200 mg/kg MT group. The expression of *Mn-SG* in the 100 and 200 mg/kg MT groups was relatively low and significantly different from that of the control and 50 mg/kg MT groups (*p* < 0.05). As shown in Figure 5, *Mn-Vg* and *Mn-Vgr* mRNA levels in females were measured at 40 days after different concentrations of MT were administered. Increases in MT concentration dramatically decreased the expression levels of *Mn-Vg* and *Mn-Vgr*. The expression levels of *Mn-Vg* and *Mn-Vgr* at 100 and 200 mg/kg MT were similar, and both were significantly different from the control (*p* < 0.05).

### 3.5. Analysis of Sex-Related Genes by qRT-PCR

Figure 6 shows *MniDMRT11E*, *MnSoxE1*, and *MnFoxl2* mRNA levels in males and females measured at 40 days after different concentrations of MT. All the samples measured were whole prawns from three separate individuals in the same group. In male prawns, the *MniDMRT11E*, *MnFoxl2*, and *MnSoxE1* expression levels of 200 mg/kg MT were 8.65-, 3.75-, and 3.45-fold greater than those of the control group (*p* < 0.05), respectively. The expression levels of these three genes in females were different from those in males. In female prawns, the expression levels of *MniDMRT11E* and *MnFoxl2* gradually decreased with the increase in MT concentration, and at 200 mg/kg MT, they were 2.56- and 3.40-fold less than those in the control group (*p* < 0.05), respectively. The *MnSoxE1* expression level at 200 mg/kg MT was not significantly different from that of the control group.

## 4. Discussion

### 4.1. Sex Reversal

This study aimed to determine whether dietary supplementation with different concentrations of MT could induce sex reversal in the oriental river prawn and whether it affected growth and gonadal development. The period before sexual differentiation is known as the period of instability and is usually considered to have a higher degree of sexual plasticity because undifferentiated gonads are more sensitive to exogenous steroids [37]. Previous studies have revealed that the period of unstable sexual differentiation in the oriental river prawn was before PL10 and that the gonads were almost fully developed at PL25 [7]. MT has been successfully used to masculinize females in *P. olivaceus* [38], *D. rerio* [39], *D. labrax* [40], *O. mykiss* [41], and so on. In this experiment, more males were observed after feeding MT to post-larval juvenile oriental river prawns, which was evidence that sex reversal was achieved by using MT as a dietary supplementation after gonadal development. Histological observations also demonstrated that spermatozoa and ovaries coexisted. These results provided an important basis for the regulation of sex differentiation with MT and the establishment of monosex cultures in juvenile oriental river prawns. However, we discovered that the sex ratio did not reach 100% male. This may require further research, such as extending the feed period or increasing the concentration of hormones. Another possible speculation is that the efficiency of sex reversal may be influenced by environmental factors, such as water temperature and climate. Activation of the stress response axis is a key process in environment-induced sex plasticity in fish [42]. Furthermore, recent genetic studies in mice suggest that gonadal sexual reversal requires active maintenance to suppress the opposite sex during adulthood [43]. Therefore, further refinement of feeding times and doses may be needed to make a single feeding sufficient to fully masculinize the population of this species.

In the oriental river prawn, the expression of androgen-related genes is decreased in AG by *IAG* RNAi [22]. In *M. rosenbergii*, fully functional sex reversal can be achieved by *IAG* silencing [6]. Interestingly, in this study, the expression of *Mn-IAG* in the 100 and 200 mg/kg MT groups was only an eighteenth of that in normal male prawns. Although *Mn-SG* declined, it remained at a certain level. Salmon (*O. kisutch*) larvae retain approximately 10% of exogenous steroids 16 days after cessation of treatment [44]. Most rats with a testis removed maintained spermatogenesis in numbers after receiving exogenous hormone treatment. These results showed that the male characteristics of neo-males were maintained by exogenous androgens and that neo-males cannot produce endogenous androgens on their own [45]. Another interesting phenomenon was that the sex ratio decreased with culture at relatively lower concentrations of 50 and 100 mg/kg MT. In sea bass, 200 or more days after MT withdrawal, under the test conditions, the induction of sterility was reversible, and it was observed that a few germ cells at 426 days post-fertilization (DPF) were able to resume mitosis [46], indicating that neo-males retain the possibility of reverting to females. However, a similar phenomenon has not been observed in crustaceans. It has been shown that endogenous hormones produced by the testis are more effective than exogenous hormones in maintaining spermatogenesis at the same concentration. Therefore, a reasonable speculation is that with the growth of prawns, their own endogenous hormones gradually take a dominant role and influence the development of their secondary sexual characteristics.

### 4.2. Gonadal Development

In aquatic species, many studies have reported that MT promoted the maturation of male organs and the formation of secondary sexual characteristics, while inhibiting ovarian development [47,48]. In studies on grass carp (*C. idella)*, it has been proposed that this phenomenon occurs because MT inhibits ovarian development and the energy used for ovarian development is used for growth and development [49]. Furthermore, a previous study on the oriental river prawn revealed that dietary supplementation with MT at concentrations greater than 500 mg/kg delayed ovarian development in juvenile prawns [50]. In this study, the WGR and SGR of female prawns in the 100 and 200 mg/kg MT were significantly increased compared to those of the control group. Meanwhile, the expression levels of *Mn-Vg* and *Mn-Vgr* were significantly lower in the experimental group than the control for females at the same developmental stage, indicating that the growth of female prawns was strengthened by inhibited ovarian development.

The WGR and SGR of male prawns decreased after MT supplementation, and no significant difference between male and female prawns occurred in the 200 mg/kg MT group. Histological observation revealed that the testis development of the experimental group was in the spermatocyte phase, and less sperm was found. It is possible that the neo-males still had female characteristics such as small body size and slow growth rate. Another suggestion is that this negative growth performance may be the result of energy consumption due to rapid gonadal development [51]. In a study on peacock fish (*P. reticulata*), high concentrations of MT also caused weight and testis growth rates to decrease dramatically in males [52]. They also caused malformation or even agenesis of the sperm duct system in *E. suillus* [53]. These results showed that over-reliance on exogenous hormones led to the failure of organs that produce sex hormones.

### 4.3. Sexual Differentiation Genes

DMRT proteins share a distinctive zinc-finger DNA-binding motif termed the DM domain [54,55]. The expression of *DMRT* is upregulated in association with androgen-induced gonadal masculinization. *DMRT* was found to be upregulated during sex reversal in rainbow trout [56]. Similarly, in this experiment, *DMRT11E* was significantly upregulated in neo-males. *DMRT11E* RNAi in *M. nipponense* showed negative regulation with the *Vg* gene and positive regulation with the *IAG* gene [57]; this result is the same as that found in this study.

*Foxl2* is a member of the forkhead/HNF-3-related family of transcription factors that plays an important role in the development and maintenance of ovarian function in vertebrates [58,59]. The significant downregulation of *Foxl2* in sex-reversed mandarin fish showed that *Foxl2* is involved in sex differentiation [9]. A previous study on RNAi of *Foxl2* in the oriental river prawn found considerable evolutionary differences in *Foxl2* in crustaceans and fish; *Mn-Foxl2* mRNA expression levels were almost 15-fold higher in testes than in ovaries [60].

*SoxE* subgroup transcription factors are thought to play essential roles in mammalian sex determination and gonadal development [61,62]. In teleost fish, they may only play a role in germline maintenance rather than in sexual determination [63]. In previous studies, *SoxE1* was found to be involved in the early differentiation of males in the oriental river prawn and expressed at various stages of gonad development [64].

In the present study, the *MniDMRT11E*, *MnFoxl2,* and *MnSoxE1* expression levels of 200 mg/kg MT were 8.65-, 3.75-, and 3.45-fold greater than those in the control group males, respectively. These expression levels were negatively correlated with *IAG*, indicating that these genes are possibly involved in the synthesis and secretion of androgens and are key genes in gonadal androgenization. In female prawns, the expression levels of *MniDMRT11E* and *MnFoxl2* in the experimental group were significantly lower than those in the control, indicating that they are not only involved in the sexual differentiation of males but also key genes for ovarian development. Meanwhile, the expression of *SoxE1* was higher at 50 and 100 mg/kg MT than in the control, but not different from the control at 200 mg/kg, suggesting that it was also involved in ovarian development but did not play a major role.

## 5. Conclusions

This present study demonstrated that vertebrate sexual hormones could induce sex reversal in crustaceans, and determined vertebrate sexual hormone (MT) concentrations and periods that induced sex reversal in the oriental river prawn. We found that MT at a high concentration (200 mg/kg) induced sex reversal. However, neo-males induced by low concentrations of MT (50 and 100 mg/kg) may revert to females. Neo-males were maintained by exogenous androgens, and over-reliance led to slow testis growth, small body size, and low growth rate, but sperm was still produced. Sex-related genes played a role in male sexual differentiation and female ovarian development. The current research provides an important theoretical basis for identifying the type of chromosomal sex determination in the oriental river prawn and for achieving a monosex culture.

## Figures and Tables

**Figure 1 animals-13-01369-f001:**
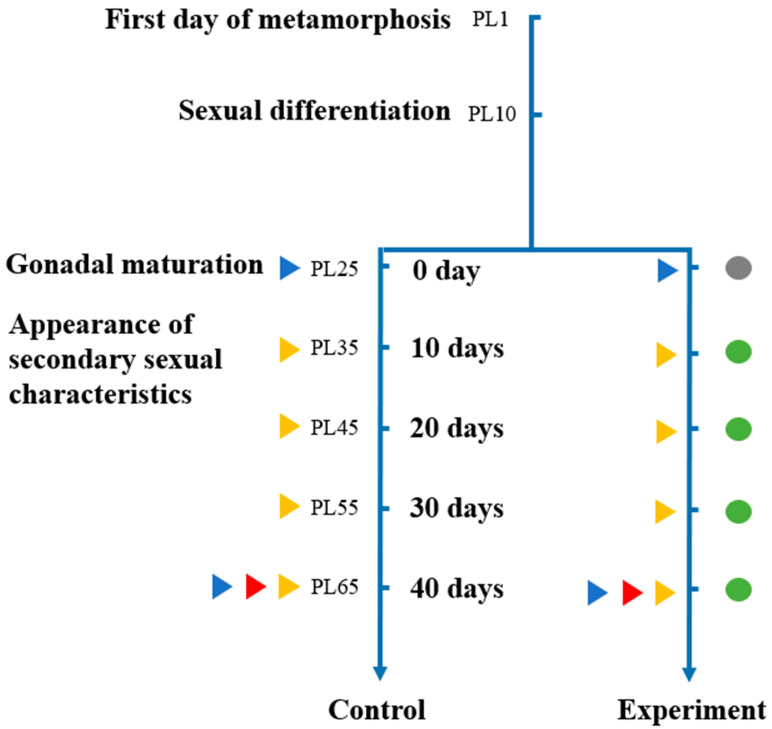
Experimental design. Yellow triangles represent the sex ratio statistic stage, red triangles represent the histological sampling stage, and blue triangles represent the growth data collection stage. The gray circle indicates that secondary sexual characteristics have not appeared and the green circles indicate that secondary sexual characteristics have appeared.

**Figure 2 animals-13-01369-f002:**
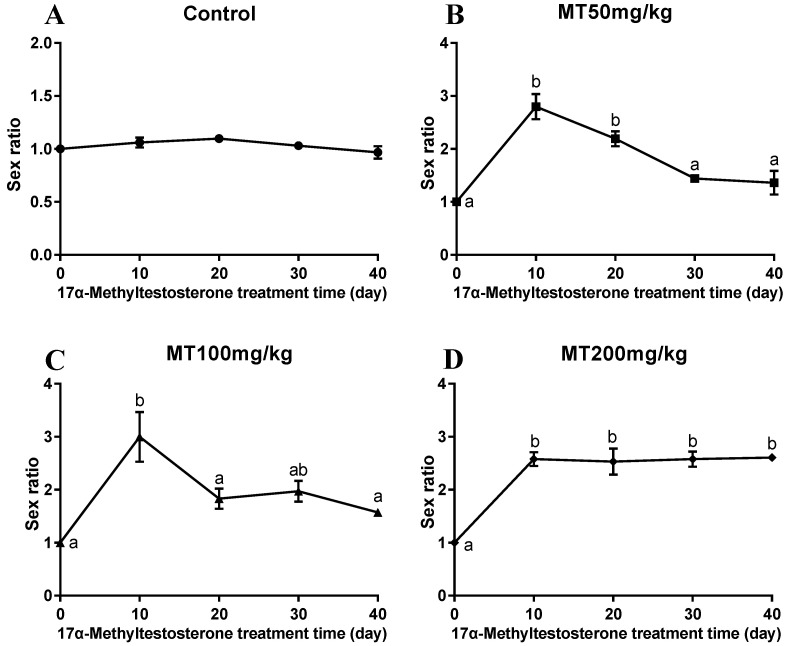
The sex ratio at four different concentrations of MT over time. (**A**) Control; (**B**) 50 mg/kg MT; (**C**) 100 mg/kg MT; (**D**) 200 mg/kg MT. Data are shown as mean ± SEM of tissues from separate individuals (n = 3). Lowercase letters indicate differences in expression between different samples in the same group.

**Figure 3 animals-13-01369-f003:**
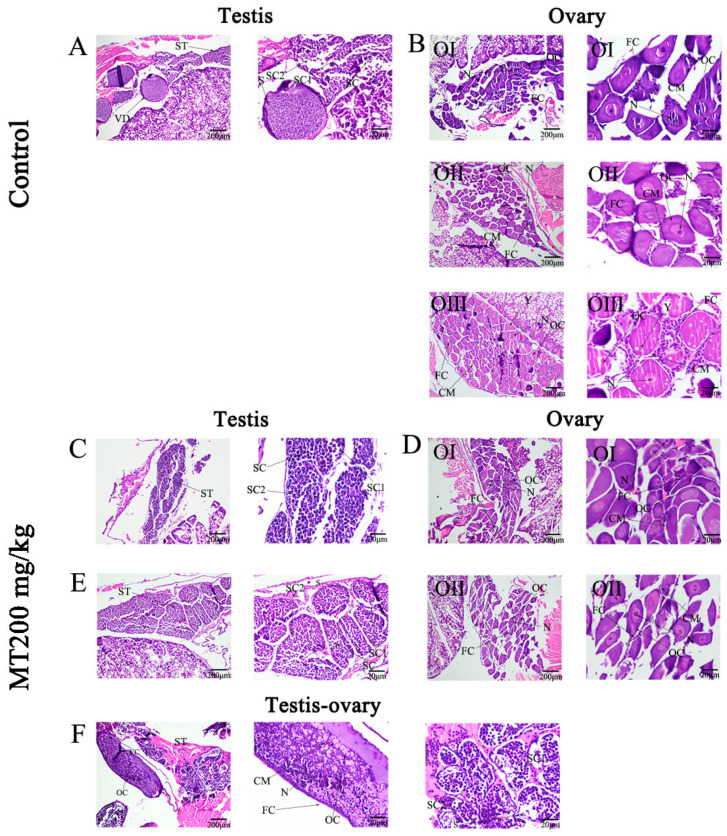
Histological sections of (**A**) male prawn testis in the control, (**B**) oocyte I, oocyte II and oocyte II of female prawns in the control, (**C**,**E**) testis of male prawns in the 200 mg/kg MT group, (**D**) oocyte I and oocyte II of female prawns in the 200 mg/kg MT group, and (**F**) testis–ovary of male prawns in the 200 mg/kg MT group. N: nucleus; ST: spermatid; SC: spermatocyte; SC1: primary spermatocyte; SC2: second spermatocyte; S: sperm; VD: vas deferens; OC: ovarian cavity; FC: follicle cells; CM: cytoplasmic membrane; Y: yolk granule. Scale bars: 200 µm and 20 µm.

**Figure 4 animals-13-01369-f004:**
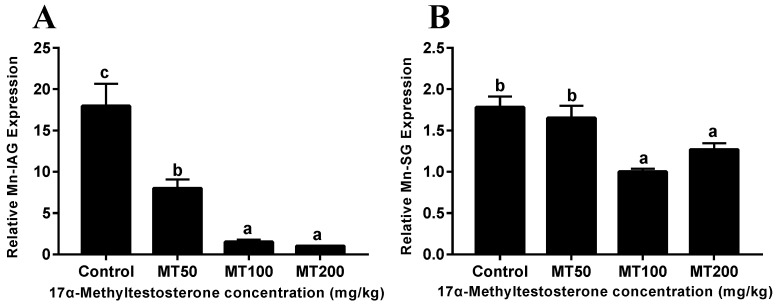
Expression characterization of *Mn-IAG* and *Mn-SG* in male prawns treated with different concentrations of MT for 40 days. (**A**) Expression levels of *Mn-IAG* in male prawns treated with different concentrations of MT for 40 days; (**B**) expression levels of *Mn-SG* in male prawns treated with different concentrations of MT for 40 days. The amount of *Mn-IAG* and *Mn-SG* mRNA was normalized to the *EIF* transcript level. Data are shown as mean ± SEM of whole prawns from separate individuals in the same group (n = 3). Lowercase letters indicate the difference in expression between different samples.

**Figure 5 animals-13-01369-f005:**
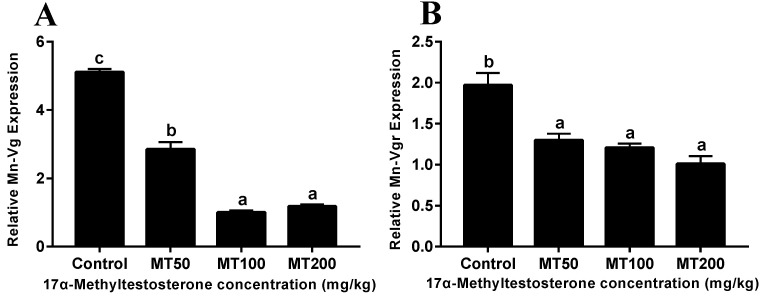
Expression characterization of *Mn-Vg* and *Mn-Vgr* in female prawns treated with different concentrations of MT for 40 days. (**A**) Expression levels of *Mn-Vg* in female prawns treated with different concentrations of MT for 40 days; (**B**) expression levels of *Mn-Vgr* in female prawns treated with different concentrations of MT for 40 days. The amount of *Mn-Vg* and *Mn-Vgr* mRNA was normalized to the *EIF* transcript level. Data are shown as mean ± SEM of whole prawns from separate individuals in the same group (n = 3). Lowercase letters indicate the difference in expression between different samples.

**Figure 6 animals-13-01369-f006:**
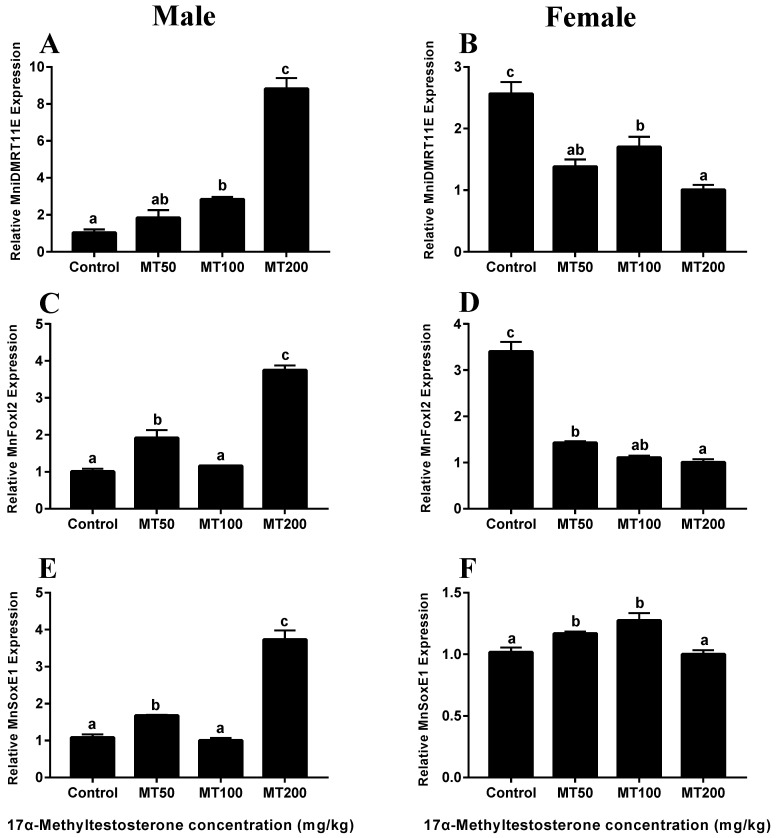
Expression characterization of *MniDMRT11E*, *MnFoxl2*, and *MnSoxE1* in male and female prawns treated with different concentrations of MT for 40 days. (**A**,**C**,**E**) Expression levels of *MniDMRT11E*, *MnSoxE1*, and *MnFoxl2* in male prawns treated with different concentrations of MT for 40 days; (**B**,**D**,**F**) expression levels of *MniDMRT11E*, *MnSoxE1*, and *MnFoxl2* in female prawns treated with different concentrations of MT for 40 days. The amount of *MniDMRT11E*, *MnFoxl2*, and *MnSoxE1* mRNA was normalized to the *EIF* transcript level. Data are shown as mean ± SEM of whole prawns from separate individuals in the same group (n = 3). Lowercase letters indicate the difference in expression between different samples.

**Table 1 animals-13-01369-t001:** The body weight of oriental river prawn at different developmental stages.

Development Days (d)	Body Weight (mg)	Average Body Weight (mg)
PL1	3.4–3.5	3.4
PL4	5.3–6.4	5.9
PL7	6.4–7.0	6.7
PL10	7.0–8.5	7.6
PL13	9.2–11.9	10.7
PL16	14.8–18.6	16.6
PL19	35.6–43.8	35.0
PL22	31.3–37.6	38.5
PL25	40.5–46.4	43.1
PL28	62.8–66.0	64.5
PL31	101.9–155.0	127.8

Note: PL: post-larvae developmental stage. The results are presented as mean ± SE (standard error of the mean).

**Table 2 animals-13-01369-t002:** Male and female appearance characteristics of oriental river prawn.

Characteristic	Male	Female
Physique	Big	Small
The second pereiopod	About 1.5 times body length	No longer than body length
Width of the fifth pereiopod	Narrower, equidistant	Greater than width of the fourth pereiopod
The second ventral extremity	With a rod-like projection	None
Genital pore	Located at the base of the fifth pereiopod	Located medial to the base of the third pereiopod

Note: The difference between male and female oriental river prawns is distinguished mainly from the physique, second pereiopod, width of the fifth pereiopod, second ventral extremity, and genital pore.

**Table 3 animals-13-01369-t003:** Primers of sequence used.

Primer Name	Sequence (5′→3′)	Description
*EIF*-F	CATGGATGTACCTGTGGTGAAAC	FWD primer for *EIF* expression
*EIF*-R	CTGTCAGCAGAAGGTCCTCATTA	RVS primer for *EIF* expression
*Vg*-F	GAAGTTAGCGGAGATCTGAGGT	FWD primer for *Vg* expression
*Vg*-R	CCTCGTTGACCAATCTTGAGAG	RVS primer for *Vg* expression
*Vgr*-F	ACCACTCGGATGAGGACGACT	FWD primer for *Vgr* expression
*Vgr*-R	CCATCTTTGCACTGGTAGTGGT	RVS primer for *Vgr* expression
*IAG*-F	CGCCTCCGTCTGCCTGAGATAC	FWD primer for *IAG* expression
*IAG*-R	CCTCCTCCTCCACCTTCAATGC	RVS primer for *IAG* expression
*SG*-F	ACCCTAGCCCCAGTACGTGTT	FWD primer for *SG* expression
*SG*-R	AGAGGTGGTGAAGCTGTCTCTCA	RVS primer for *SG* expression
*DMRT*-F	ACGACCTTAGTAGGATGGACAGT	FWD primer for *DMRT* expression
*DMRT*-R	GAGTGGAGGCAATAGAATGGGTA	RVS primer for *DMRT* expression
*Foxl2*-F	AAATCCCTGTACGACCACATCG	FWD primer for *Foxl2* expression
*Foxl2*-R	CTTCGTCGGGTAGAGCATCTCC	RVS primer for *Foxl2* expression
*SoxE*-F	ACATAGATCGCGCAGAAATGAAC	FWD primer for *SoxE* expression
*SoxE*-R	CCAAGGAAGGAAGACTTGTGAGT	RVS primer for *SoxE* expression

**Table 4 animals-13-01369-t004:** Effects of different concentrations of 17α-methyltestosterone on growth performance of juvenile prawns.

Sex	Index	Group (mg/kg)				Pr > F ^1^		
		Control	MT50	MT100	MT200	ANOVA	Linear Trend	Quadratic Trend
Male	IMW (mg)	44.0 ± 0.3	43.9 ± 0.3	43.0 ± 0.2	42.8 ± 0.4	0.02	0.00	0.96
	FMW (mg)	965.8 ± 24.0 ^b^	807.7 ± 55.9 ^a^	791.6 ± 37.0 ^a^	736.8 ± 37.7 ^a^	0.00	0.00	0.21
	WGR (%/g)	2092.47 ± 47.44 ^b^	1736.74 ± 122.33 ^a^	1741.13 ± 81.56 ^a^	1628.87 ± 77.43 ^a^	0.00	0.00	0.17
	SGR (%/d)	6.17 ± 0.04 ^b^	5.78 ± 0.13 ^a^	5.811 ± 0.09 ^a^	5.86 ± 0.05 ^a^	0.00	0.00	0.17
Female	IMW (mg)	44.0 ± 0.3	43.9 ± 0.3	43.0 ± 0.2	42.8 ± 0.4	0.02	0.00	0.96
	FMW (mg)	546.4 ± 20.8 ^a^	573.1 ± 32.1 ^a^	634.4 ± 27.5 ^ab^	706.0 ± 30.0 ^b^	0.00	0.00	0.43
	WGR (%/g)	1141.05 ± 47.35 ^a^	1204.61 ± 71.71 ^a^	1377.84 ± 67.14 ^ab^	1549.79 ± 72.32 ^b^	0.00	0.00	0.41
	SGR (%/d)	5.03 ± 0.07 ^a^	5.11 ± 0.11 ^a^	5.37 ± 0.09 ^ab^	5.59 ± 0.09 ^b^	0.00	0.00	0.46

Note: The data are expressed as mean ± SEM (n = 150). The lowercase letters indicate significant differences among different MT concentrations (*p* < 0.05). IMW: initial mean weight. FMW: final mean weight. WGR: weight growth rate. SGR: specific growth rate. ^1^ Significance probability associated with the F-statistic.

## Data Availability

The data presented in this study are available on request from the corresponding author for scientific purposes.

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
