# Peer review of "Sex Reversal Induced by Dietary Supplementation with 17α-Methyltestosterone during the Critical Period of Sex Differentiation in Oriental River Prawn (Macrobrachium nipponense)"

_animals, 2023, doi:10.3390/ani13081369_

Round 1

Author Response

Dear Reviewers:

We thank the reviewers for their time and effort spent to critically review our manuscript. Based on these comments and suggestions, we have made careful modifications on the manuscript. All the changes made to the text are in purple or blue color. Below, we attached a point by point response to all questions and concerns.

Responses to the Comments from Reviewer 1

Abstract 

  1. This manuscript contains interesting results concerning increase of male sex ratio (ca. 3-fold) by synthetic androgen exposure. The authors need to discuss why androgen does not induce 100% male in Discussion.

Response: Thanks for reviewer’s comment. We have already discussed " why androgen does not induce 100% male " (line 334-342).

  1. Figure 3: need larger seize photos; Minor editorial revisions will be needed

Response: Thanks for reviewer’s comment. Figure 3 has been changed to 600dpi. (line 236)

  1. Line 21: add groups after experimental

Response: Thanks for reviewer’s comment. We have added “groups” after “experimental”. (line 22)

  1. Line 25: “Neo-males” should be appropriately explained

Response: Thanks for reviewer’s comment. We have explained “Neo-males”. (line 26)

  1. Lines 36, 39: add reference(s) after female, differentiation, respectively

Response: Thanks for reviewer’s comment. We have added reference(s) after female, differentiation, respectively. (line 38,41)

  1. Line 61: Epinephlus should be E. (italic)

Response: Thanks for reviewer’s comment. We have modified it. (line 61)

  1. Line 68: Chinese should be Chinese

Response: Thanks for reviewer’s comment. We have modified it. (line 69)

  1. Line 70: Macrobrachium Rosenbergii (italic) should be M. rosenbergi (italic)

Response: Thanks for reviewer’s comment. We have modified it. (line 71)

  1. Line 75: spell out SG

Response: Thanks for reviewer’s comment. We have spell out SG. (line 75)

  1. Line 79: spell out IAG

Response: Thanks for reviewer’s comment. We have spell out IAG. (line 80)

  1. Line 82: add (RNAi) after knockdown

Response: Thanks for reviewer’s comment. We have added “RNAi” after “knockdown”. (line 83)

  1. Lines 85, 86: Vg (italic) should be VG

Response: Thanks for reviewer’s comment. We have modified it. (line 84-87)

  1. Line 88: VG (italic) should be Vg (italic)

Response: Thanks for reviewer’s comment. We have modified it. (line 89)

  1. Line 89: add space after ovary

Response: Thanks for reviewer’s comment. We have added “space” after “ovary”. (line 90)

  1. Table 1: Body weight and Average body weight should be (mg) which is easier to read the Talbe

Response: Thanks for reviewer’s comment. Table 1 has modified it. (line 113)

  1. Line 113: add (city, China) after Ltd.

Response: Thanks for reviewer’s comment. We have added “city, China” after “Ltd”. (line 117)

  1. Line 116: add citation after hormones

Response: Thanks for reviewer’s comment. We have added citation after “hormones

”. (line 119)

  1. Line 116: add (CAS number and purity) after MT

Response: Thanks for reviewer’s comment. We have added “CAS number and purity” after “MT”. (line 119)

  1. Line 116: add (Beijing, China) after Ltd.

Response: Thanks for reviewer’s comment. We have added “Beijing, China” after Ltd. (line 120)

  1. Lines 121, 122: minutes should be min

Response: Thanks for reviewer’s comment. We have modified it. (line 126)

  1. Line 128: add space after 2.3.

Response: Thanks for reviewer’s comment. We haves added. (line 132)

  1. Line 131: delete days after 10, 20, 30

Response: Thanks for reviewer’s comment. We have modified it. (line 136)

  1. Line 139: delete days after 10, 20, 30 add cmmas before and after respectively

Response: Thanks for reviewer’s comment. We have modified it. (line 144)

  1. Line 183: add city, states after Bio-Rad,

Response: Thanks for reviewer’s comment. We have modified it. (line 190)

  1. Line 199: Resut should be Results ? follow the journal style

Response: Thanks for reviewer’s comment. We have modified it. (line 209)

  1. Line 202 and throughout the text: delete period after Figure

Response: Thanks for reviewer’s comment. We have modified it. (line 211)

  1. Line 202-203: improve the sentence

Response: Thanks for reviewer’s comment. We have modified it. (line 211)

  1. Line 206: significant should be significantly delete days after 10                            20 day should be 20 days    delete days after

Response: Thanks for reviewer’s comment. We have modified it. (line 215)

  1. Line 210: add (Figure 2D after groups

Response: Thanks for reviewer’s comment. We have added “Figure 2D” after groups. (line 218)

  1. Line 213: this sentence should be revised

Response: Thanks for reviewer’s comment. We have revised it. (line 222)

  1. Line 216: add number of replicates as n=

Response: Thanks for reviewer’s comment. We have modified it. (line 224)

  1. Line 238: (Tables 4 and 5)

Response: Thanks for reviewer’s comment. We've merged the two tables. (line 262)

  1. Line 239: weight growth rate (WGR) and specific growth rate (SGR)

Response: Thanks for reviewer’s comment. The WGR and SGR have already been defined. (line 155)

  1. Line 240: delete mg/kg after MT50, MT100

Response: Thanks for reviewer’s comment. We have deleted mg/kg after MT50, MT100. (line 247)

  1. Line 265 and throughout the text: folder should be fold

Response: Thanks for reviewer’s comment. We have modified it. (line 274)

  1. Lines 279, 286, 306: add number of samples used as n=

Response: Thanks for reviewer’s comment. We have modified it. (line 288,296,316)

  1. Line 308: add space after 4.1.

Response: Thanks for reviewer’s comment. We have added space after 4.1. (line 319)

  1. Line 323: RNA interference-induced knockdown of IAG should be IAG RNAi

Response: Thanks for reviewer’s comment. We have modified it. (line 344)

  1. Line 325: delete mg/kg after MT100

Response: Thanks for reviewer’s comment. We have deleted mg/kg after MT100. (line 345)

  1. Line 328: add reference(s) after their own

Response: Thanks for reviewer’s comment. We have added reference(s) after “their own”. (line 352)

  1. Line 335: add space after 4.2.

Response: Thanks for reviewer’s comment. We have added space after 4.1. (line 363)

  1. Line 338: add scientific name after grass carp

Response: Thanks for reviewer’s comment. We have added scientific name after grass carp. (line 366)

  1. Line 341: supple mentation should be supplementation

Response: Thanks for reviewer’s comment. We have changed it. (line 369)

  1. Line 343: delete mg/kg after MT100line 356: Epinephelus (italic) should be E. (italic)

Response: Thanks for reviewer’s comment. We have modified it. (line 371 385)

  1. Line 359: add space after 4.3.

Response: Thanks for reviewer’s comment. We have added space after 4.3. (line 387)

  1. Line 374: add comma after fish

Response: Thanks for reviewer’s comment. We have added comma after fish. (line 403)

  1. Line 379: …3.45 fold greater, respectively, than…

Response: Thanks for reviewer’s comment. We have revised this sentence. (line 408)

  1. Line 389: …vertebrate sexual hormones… should be …a vertebrate sexual hormone (MT) ….

Response: Thanks for reviewer’s comment. We have revised this sentence. (line 418)

  1. Line 395: …female promote.. should be …female…

Response: Thanks for reviewer’s comment. We have revised this sentence. (line 424)

  1. References: need to follow the journal style: journal names should be appropriately abbreviated, scientific names of animals should be italic, titles should not be large capital except the first letter of the first word

Line 471: Prod. Natl. Adad. Sci. USA

Line 566: journal name, volume, pages are missing

Line 571: PLoS One

Response: Thanks for reviewer’s comment. We have modified the format of the references and updated it to new studys. (line 452-609)

Reviewer 2 Report

The authors investigated the effect of Sex reversal induced by dietary supplementation with 17α-Methyltestosterone during the critical period of sex differentiation in oriental river prawn Macrobrachium nipponense). This manuscript (MS) was clearly written and easy to understand. This work can help the sustainability of this species farming as few studies have been done on this topic. However, some major issues significantly compromised the quality of this MS.

First, the manuscript needs to be edited by a native English speaker to improve the language of the MS and fix errors.

However, I have touched on some more points that can contribute to the improvement of this MS.

·       Line 14-27, please make sure you change the style of the abstract based on the Animals journal.

·       Line 75 and other parts, please make sure you defined the abbreviations for the first time in the MS.

·       Line 99, how did you know they were healthy?

·       Table, please provide based on mg as well.

·       Line 114, some of these ingredients have it!! How do you know they do not have it? You had to analyse it.

·       Line 124, how did it take?

·       Table 2, please tidy up this table, and justify it.

·       Line 196, please use the Tukey test and update the results and discussion.

·       Figures and tables, if there is no significant difference, please delete subsets (a, b, c).

·       Figure 3, please make them bigger to be more clear to see.

·       Table 4, please change them to mg and maximum two-three decimals.

·       Please merge tables 4 and 5.

·       The statistical analysis should be changed to regression to see whether there is a polynomial or/and linear regression to eventually provide an optimum level. However, they can report both ANOVA and Regression if it was some parameters that using ANOVA for them make more sense. Please check this paper to see how you can report the results. https://www.sciencedirect.com/science/article/pii/S0044848619305861   

·       Here and elsewhere, report P uppercase and italic (P<0.05).

·       Throughout the MS, if there is no significant difference, no need to report the P-value.

·       Please reorder the keywords alphabetically and capitalise each word.

·       Please write the abstract more numerically about the results. You can do it by adding their numbers in parentheses.

·       Here and throughout the MS, please first mention the common name plus the scientific name, and for the rest of the MS, just report the common name.

·       Please update the introduction with recent works, as many studies are available from the last two years, which were not included in this section.

·       Please mention the novelty of your work in the last paragraph of the introduction.

·       Some parts of the discussion are better updated with research in 2022 and 2023 as they refer to some old references. Please update the discussion with the latest studies as much as possible.

·       Although you wrote this section well, you can still improve it by answering these questions and annotating them into the discussion section. Why were these results observed? Discuss more possible reasons.

Tables and Figures

•            Please explain a little bit about your experimental diets per each Table and Figure. Each table and figure should represent enough information separately from the text.

•            Double-check the units and titles of all Tables.

•            Please mention in the footnote of all Tables which kind of statistical method you used for comparing the means.

When revising your manuscript, please consider all issues mentioned in the reviewers' comments carefully with clear outlines for every change made in response to their comments including suitable rebuttals for any comments you deem inappropriate. Please itemise your response to each review comment, and highlight the revised at re-submission.

Best regards

Author Response

Dear Reviewer:

We thank the reviewer for their time and effort spent to critically review our manuscript. Based on these comments and suggestions, we have made careful modifications on the manuscript.  Below, we attached a point by point response to all questions and concerns.

Responses to the Comments from Reviewer 2

The authors investigated the effect of Sex reversal induced by dietary supplementation with 17α-Methyltestosterone during the critical period of sex differentiation in oriental river prawn (Macrobrachium nipponense). This manuscript (MS) was clearly written and easy to understand. This work can help the sustainability of this species farming as few studies have been done on this topic. However, some major issues significantly compromised the quality of this MS.

1. First, the manuscript needs to be edited by a native English speaker to improve the language of the MS and fix errors.

Response: Thanks for reviewer’s comment. This manuscript has been edited by a native English speaker. We hope that the revision will be able to reach the publication level of the journal.

  1. However, I have touched on some more points that can contribute to the improvement of this MS.

Line 14-27, please make sure you change the style of the abstract based on the Animals journal.

Response: Thanks for reviewer’s comment. The style of the abstract has changed to the Animals journal. (line 14-29)

  1. Line 75 and other parts, please make sure you defined the abbreviations for the first time in the MS.

Response: Thanks for reviewer’s comment. We have defined the abbreviations when it first appeared in the manuscript. (line 53-76)

  1. Line 99, how did you know they were healthy?

Response: Thanks for reviewer’s comment. The criteria for selecting female prawns are big size, good vitality, normal feeding, yellow-green body color, and have green eggs.

  1. Table, please provide based on mg as well.

Response: Thanks for reviewer’s comment. We have modified it. (line 131 260)

  1. Line 114, some of these ingredients have it!! How do you know they do not have it? You had to analyse it.

Response: Thanks for reviewer’s comment. This sentence is intended to express that the dietary we used does not contain any other hormones. We have deleted it because of ambiguities arising from the expression.

  1. Line 124, how did it take?

Response: Thanks for reviewer’s comment. This sentence is changed to “The diets were left until the color was the same as normal diets and without the smell of alcohol”. (line 128) When there is no irritating alcohol taste in the dietary and it does not affect the feeding, we keep it in reserve.

  1. Table 2, please tidy up this table, and justify it.

Response: Thanks for reviewer’s comment. We have added references after “table 2”. (line 146)

  1. Line 196, please use the Tukey test and update the results and discussion.

Response: Thanks for reviewer’s comment. We have used Tukey test and update the results. The results are basically the same as the original. (line 206)

  1. Figures and tables, if there is no significant difference, please delete subsets (a, b, c).

Response: Thanks for reviewer’s comment. We have modified it. (line 222 262)

  1. Figure 3, please make them bigger to be more clear to see.

Response: Thanks for reviewer’s comment. Figure 3 has been changed to 600dpi. (line 236)

11.Table 4, please change them to mg and maximum two-three decimals.

Response: Thanks for reviewer’s comment. We have modified it. (line 262)

12.Please merge tables 4 and 5.

Response: Thanks for reviewer’s comment. Tables 4 and 5 have been merged. (line 262)

  1. The statistical analysis should be changed to regression to see whether there is a polynomial or/and linear regression to eventually provide an optimum level. However, they can report both ANOVA and Regression if it was some parameters that using ANOVA for them make more sense. Please check this paper to see how you can report the results. https://www.sciencedirect.com/science/article/pii/S0044848619305861   

Response: Thanks for reviewer’s comment. According to the article you provided, we performed a linear regression analysis of IMW, FMW, SGR and WGR for males and females. (line 262) The results showed that the linear relation was very significant after feeding MT.

  1. Here and elsewhere, report P uppercase and italic (P<0.05).

Response: Thanks for reviewer’s comment. We have modified it in manuscript.

  1. Throughout the MS, if there is no significant difference, no need to report the P-value.

Response: Thanks for reviewer’s comment. We have modified it in manuscript.

  1. Please reorder the keywords alphabetically and capitalise each word.

Response: Thanks for reviewer’s comment. We have reordered the keywords alphabetically and capitalized the first letter. (line 28)

  1. Please write the abstract more numerically about the results. You can do it by adding their numbers in parentheses.

Response: Thanks for reviewer’s comment. We have added some numbers to the abstract and used them to represent the results.

  1. Here and throughout the MS, please first mention the common name plus the scientific name, and for the rest of the MS, just report the common name.

Response: Thanks for reviewer’s comment. We have modified “M. nipponense” to “oriental river prawn” in manuscript.

  1. Please update the introduction with recent works, as many studies are available from the last two years, which were not included in this section.

Response: Thanks for reviewer’s comment. We have updated the references in the “introduction” to the recent years of research.

  1. Please mention the novelty of your work in the last paragraph of the introduction.

Response: Thanks for reviewer’s comment. We have mentioned the novelty of the work in the last paragraph of the introduction. (line 96-100)

  1. Some parts of the discussion are better updated with research in 2022 and 2023 as they refer to some old references. Please update the discussion with the latest studies as much as possible.

Response: Thanks for reviewer’s comment. We have updated the references in the manuscript to the recent years of research.

  1. Although you wrote this section well, you can still improve it by answering these questions and annotating them into the discussion section. Why were these results observed? Discuss more possible reasons.

Response: Thanks for reviewer’s comment. We have updated the references in the manuscript to the recent years of research. We discussed more questions such as "why the induction rate is not 100%" (line 337-342) and "why do males revert to females". (line 358-362)

Tables and Figures

  1. Please explain a little bit about your experimental diets per each Table and Figure. Each table and figure should represent enough information separately from the text.

Response: Thanks for reviewer’s comment. We have checked all Tables and Figures, ensure they should represent enough information separately from the text. The "experimental diets" are described in Materials and Methods.(line 111)

  1. Double-check the units and titles of all Tables.

Response: Thanks for reviewer’s comment. We have checked the units and titles of all Tables.

  1. Please mention in the footnote of all Tables which kind of statistical method you used for comparing the means.

 Response: Thanks for reviewer’s comment. We have added statistical method in the footnote of all Tables.

Round 2

Reviewer 2 Report

The authors have well improved the MS and I suggest this MS for publication. However, please read the MS in a clear version (not track change) and fix some language errors.